

# Classification of the glioma grading using radiomics analysis

Hwan-ho Cho[1,2], Seung-hak Lee[1,2], Jonghoon Kim[1,2] and Hyunjin Park[2,3]

[1] Department of Electronic and Computer Engineering, Sungkyunkwan University, Suwon, Korea
[2] Center for Neuroscience Imaging Research, Institute for Basic Science, Suwon, Korea
[3] School of Electronic and Electrical Engineering, Sungkyunkwan University, Suwon, Korea

## ABSTRACT

**Background.** Grading of gliomas is critical information related to prognosis and survival. We aimed to apply a radiomics approach using various machine learning classifiers to determine the glioma grading.

**Methods.** We considered 285 (high grade $n = 210$, low grade $n = 75$) cases obtained from the Brain Tumor Segmentation 2017 Challenge. Manual annotations of enhancing tumors, non-enhancing tumors, necrosis, and edema were provided by the database. Each case was multi-modal with T1-weighted, T1-contrast enhanced, T2-weighted, and FLAIR images. A five-fold cross validation was adopted to separate the training and test data. A total of 468 radiomics features were calculated for three types of regions of interest. The minimum redundancy maximum relevance algorithm was used to select features useful for classifying glioma grades in the training cohort. The selected features were used to build three classifier models of logistics, support vector machines, and random forest classifiers. The classification performance of the models was measured in the training cohort using accuracy, sensitivity, specificity, and area under the curve (AUC) of the receiver operating characteristic curve. The trained classifier models were applied to the test cohort.

**Results.** Five significant features were selected for the machine learning classifiers and the three classifiers showed an average AUC of 0.9400 for training cohorts and 0.9030 (logistic regression 0.9010, support vector machine 0.8866, and random forest 0.9213) for test cohorts.

**Discussion.** Glioma grading could be accurately determined using machine learning and feature selection techniques in conjunction with a radiomics approach. The results of our study might contribute to high-throughput computer aided diagnosis system for gliomas.

Corresponding author
Hyunjin Park, hyunjinp@skku.edu

## INTRODUCTION

Gliomas are primary brain tumors arising from glial cells. The grades of gliomas have been determined based on histology according to the World Health Organization standard (*Louis et al., 2007*). Recently, revised criteria have been introduced that consider genetic factors such as isocitrate dehydrogenase mutation and 1p/19q codeletion (*Louis et al., 2016*). The grading of gliomas is critical information related to prognosis and survival (*Wu et al.,*
*2010*; *Louis et al., 2016*). A scheme that dichotomizes the graded gliomas into high-grade gliomas (HGG) and low-grade gliomas (LGG) has been widely adopted. It is important to differentiate HGG from LGG for assessing tumor progression and therapy planning (*Louis et al., 2007*). An experienced observer can differentiate between the two grades well based on tumor enhancement, but a computer algorithm might match the performance of the human expert with increased speed. More importantly, the computer algorithm might contribute to developing high-throughput computer aided diagnosis system.

An algorithm known as radiomics has recently emerged as a powerful methodology to quantify the characteristics of tumors in a non-invasive manner (*Yip & Aerts, 2016*). Many studies have demonstrated that distinct tumor types in many organs can be quantified by radiomics analysis and the results of the radiomics can be used as imaging biomarkers for supporting clinical decision making (*Zacharaki et al., 2009*; *Aerts et al., 2014*; *Kickingereder et al., 2016*; *Li et al., 2016*; *Bowen et al., 2017*). Radiomics can also reveal novel characteristics of brain tumors, as demonstrated by a recent study (*Zhou et al., 2017b*). Many studies predicted the chemotherapeutic response and survival of patients with glioblastoma using a large number of imaging features based on MR imaging (*Itakura et al., 2015*; *Cui et al., 2016*; *Kickingereder et al., 2016*; *Prasanna et al., 2016*; *Lao et al., 2017*; *Zhou et al., 2017a*). Other studies have predicted prognosis using features obtained from functional imaging (*Ryu et al., 2014*; *Lee et al., 2016*). Recently, radiomics has been combined with genomics to leverage two distinct types of information to better study various tumor types (*Gutman et al., 2015*; *Li et al., 2016*; *Beig et al., 2018*; *Zinn et al., 2018*). The new approach is referred to as radiogenomics and has to potential to reveal novel findings combining two distinct high dimensional information of gene and imaging information. Many existing brain tumor studies related to radiomics mainly focused on glioblastoma which is the most aggressive glioma and considered a limited number of imaging modalities (*Ryu et al., 2014*; *Itakura et al., 2015*; *Cui et al., 2016*; *Lee et al., 2016*). Multi-modal data are high-dimensional by nature and thus handling them properly requires carefully chosen machine learning approaches. However, existing literature of radiomics using multi-modal data and various machine learning approaches is relatively scarce.

In this paper, we applied a radiomics approach combined with various machine learning approaches to multi-modal imaging of glioma patients to study whether the grade of glioma can be determined noninvasively. The aim of this study was to quantify glioma with a radiomics approach and to use the results to classify the gliomas as HGG or LGG. We used annotated multi-modal MRI imaging data from a research database (*Menze et al., 2015*; *Bakas et al., 2017a*; *Bakas et al., 2017b*; *Bakas et al., 2017c*). A total of 468 quantitative radiomics were computed for four MRI modalities and three regions of interest (ROIs). Significant features were selected using relevance and redundancy criteria. We aimed to demonstrate the effectiveness of these features on the classification of each glioma's histopathological grade using three machine learning classifiers. The overall workflow of this paper is shown in Fig. 1.

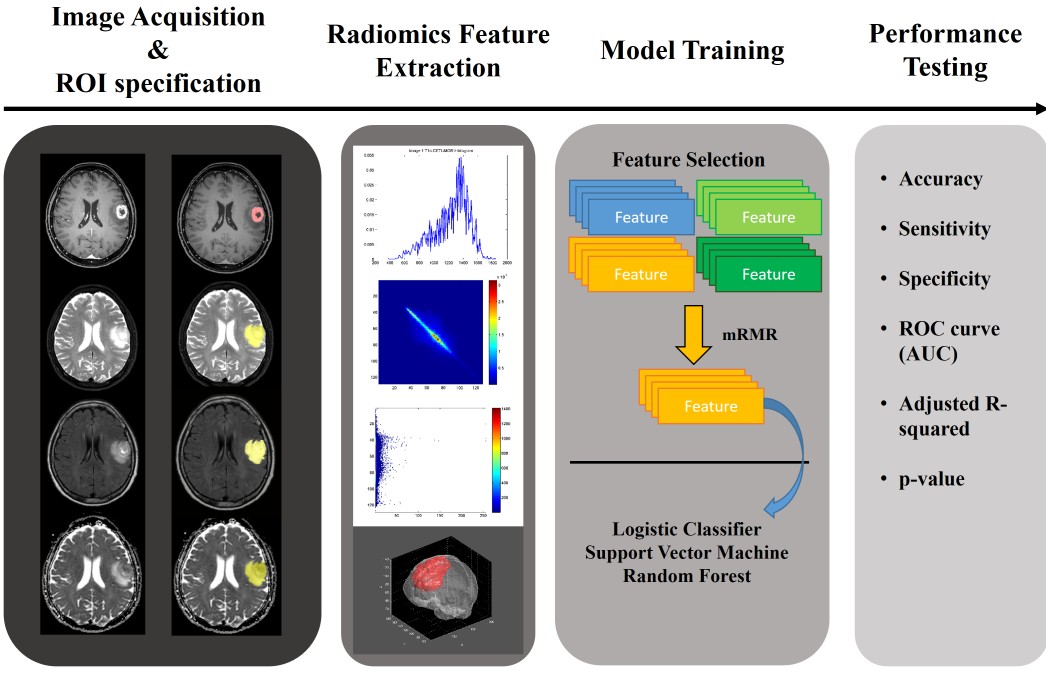

**Figure 1** Overall workflow of the study.

## MATERIALS AND METHODS

### Patients and imaging

The institutional review broad (IRB) of Sungkyunkwan University approved our study (IRB# 2015-09-007). Consent was waived for this retrospective study. Our study was performed in full accordance with local IRB guidelines. We considered data from the MICCAI Brain Tumor Segmentation 2017 Challenge (BraTS 2017) (*Menze et al., 2015*; *Bakas et al., 2017a*; *Bakas et al., 2017b*; *Bakas et al., 2017c*). This dataset was derived from pre-operative baseline scans from two variants of the Cancer Imaging Archive (TCIA) (*Clark et al., 2013*), the TCIA-glioblastoma (GBM) and TCIA-LGG collections (*Pedano et al., 2016*; *Scarpace et al., 2016*). Each is a multi-institutional data mix from eight and five institutions, respectively. Detailed patient and scanner information can be found in the data citation (*Bakas et al., 2017a*; *Bakas et al., 2017c*; *Bakas et al., 2017b*). We considered 210 HGG and 75 LGG patients. HGG included glioblastoma multi-forme (GBM) and LGG included astrocytomas, oligodendroglioma, and oligoastrocytomas (*Pedano et al., 2016*; *Scarpace et al., 2016*; *Bakas et al., 2017a*).

Each patient had pre-operative images in four modalities (T1, T1-contrast enhanced, T2, FLAIR). All images were preprocessed using the FMRIB Software Library (FSL). Each image was registered onto the common space (*Rohlfing et al., 2010*) and interpolated to a $1 \times 1 \times 1$ mm isotropic voxel grid. In addition, manual segmentations of enhancing tumors, non-enhancing tumors, necrosis, and edema in each image were also provided by the challenge organizers (*Bakas et al., 2017a*; *Bakas et al., 2017c*; *Bakas et al., 2017b*). Manual segmentation was performed using a semi-automatic method with expert

**Table 1** Institutional information of patients (*Bakas et al., 2017a*).

| Collection | Institutions | TCGA ID |
| --- | --- | --- |
| TCGA-GBM | Henry Ford Hospital, Detroit, MI | TCGA-06 |
| | CWRU School of Medicine, Cleveland, OH | TCGA-19 |
| | University of California, San Francisco, CA | TCGA-08 |
| | Emory University, Atlanta, GA | TCGA-14 |
| | MD Anderson Cancer Center, Houston, TX | TCGA-02 |
| | Duke University School of Medicine, Durham, NC | TCGA-12 |
| | Thomas Jefferson University, Philadelphia, PA | TCGA-76 |
| | Fondazione IRCCSInstituto Neuroligico C. Besta, Milan, Italy | TCGA-27 |
| TCGA-LGG | St Joseph Hospital/Medical Center, Phoenix, AZ | TCGA-HT |
| | Henry Ford Hospital, Detroit, MI | TCGA-DU |
| | Case Western Reserve University, Cleveland, OH | TCGA-FG |
| | Thomas Jefferson University, Philadelphia, PA | TCGA-CS |
| | University of North Carolina, Chapel Hill, NC | TCGA-EZ |

**Notes.**
TCGA, The Tumor Genome Atlas.

confirmation (*Bakas et al., 2017a*). These specific preprocessing choices were made by the BraTS organizational committee (*Menze et al., 2015*; *Bakas et al., 2017a*). Our study was a single source study (just from the BraTS database) and thus we adopted a five-fold cross validation to separate the training and test cohorts to reduce overfitting. Each fold had a similar ratio of HGG and LGG. The ratio of HGG and LGG was maintained between the training and test sets. Table 1 contains institutional information for all patients.

## Tumor regions of interest

We combined the three manual segmentation results provided by BraTS into ROIs to extract multi-regional radiomics features. We intended to obtain information from multiple tissue types rather than single tissue type (*Zacharaki et al., 2009*). The first region (ROI type I) was created by merging the non-enhancing tumor and necrotic region and the second region (ROI type II) was created by adding the tumor region with enhancement to the first region. The third region (ROI type III) combined the second region with the area of edema. The first region is the smallest region and the third region is the largest region inclusive of multiple compartments. Representative ROIs are shown in Fig. 2.

## Radiomics features

To compute high-dimensional imaging information needed for the radiomics approach, imaging features were calculated using all three ROIs in four modalities from 3D volume. The features were computed using a combination of open source code (*Van Griethuysen et al., 2017*) and in-house generated computer code implemented in MATLAB (Mathworks Inc. Natick, MA, USA). For most features, we used the open source software PyRadiomics so that the results could be easily reproduced. For the features not available in PyRadiomics, we used our in-house MATLAB code which is provided as a supplement material. We computed a total of 468 radiomics features per patient. We computed 24 shape-based

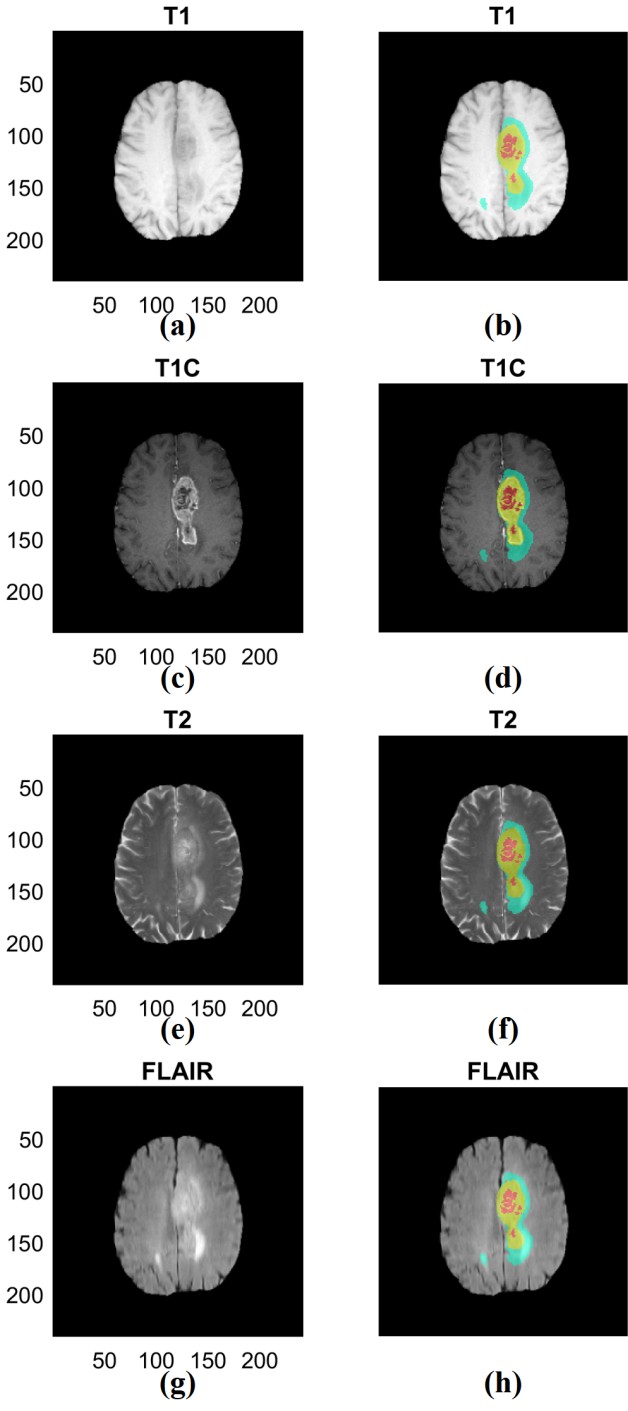

**Figure 2  Examples of three types of ROIs used in our study.** (A) T1 data. (B) ROI associated with T1. (C) T1C data. (D) ROI associated with T1C. (E) T2 data. (F) ROI associated with T2. (G) FLAIR data. (H) ROI associated with FLAIR. The left column (A) (C) (E) (G) shows different imaging modalities. The right column (B) (D) (F) (H) shows associated ROIs. The ROIs were specified in 3D but 2D representative examples are given. ROIs are visualized in the right column. Red indicates non-enhancing tumor and necrosis (ROI type I), yellow indicates enhancing tumor (ROI type II) and blue indicates edema (ROI type III) in the right column. T1; T1-weighted image, T2; T1C; T1-contrast enhanced; T2-weighted image, FLAIR; Fluid-Attenuated Inversion Recovery.

(eight for each ROI), 228 histogram-based, and 216 texture-based (192 gray-level co-occurrence matrix [GLCM] based and 24 intensity size-zone [ISZ] matrix-based) features quantifying different characteristics of the tumor (*Haralick, Shanmugam & Dinstein, 1973*; *Tixier et al., 2011*; *Davnall et al., 2012*; *Grove et al., 2015*). The histogram-based features were computed from 128 bin histogram computed over the whole intensity range. For the GLCM features, we binned intensities with 128 bins. A total of 26 matrices corresponding to 26 3D directions with offset one were computed and then averaged to yield one matrix. The averaged matrix was used to compute GLCM features. For the ISZM features, we constructed a $128 \times 256$ matrix where the first dimension was binned intensity and the second dimension was size. The size was not quantized and if a blob was larger than 256 voxels, it was considered as a blob with size 256. We considered six neighbors (four in-plane and two out-of-plane ones) for defining the size of the blob. More details can be found in the supplement.

## Feature selection

Feature values of the training cohort were normalized to $z$-scores for each feature across subjects. Different radiomics features have different units and range. Some features were designed to fall between 0 and 1, while others have a very large range. All the features were subjected to the feature selection procedure. Without feature normalization, some features might be assigned a larger weight, while others might be assigned a lower weight depending on the distribution of feature values during the feature selection. Thus, we applied $z$-score normalization to the feature values, making the range of each feature relatively uniform. A similar approach can be found in another work (*Kickingereder et al., 2016*). We selected image features which could distinguish between HGG and LGG using the minimum redundancy maximum relevance (mRMR) algorithm (*Peng, Long & Ding, 2005*). The mRMR is described with three equations. The first Eq. (1) searches for a set of features that maximizes the relevance ($D$), where $S$ is the total feature set, $c$ is the grade of glioma, $x_i$ is the individual feature, and $I$ is the information measure. The second Eq. (2) searches for a set of features that minimizes redundancy ($R$) among the selected features, where $x_i, x_j$ are the individual features. Equation (2) suppresses overlapping information among the selected features. The previous steps are combined into the third Eq. (3), where the mRMR algorithm identifies the optimal set of features that encourage maximum relevance and minimum redundancy. We performed mRMR with mutual information as the information measure and selected the top five features. We chose to select five features because we sought a compact model to distinguish between HGG and LGG images.

$$\max D(S,c), D = \frac{1}{|S|} \sum_{x_i \in S} I(x_i; c). \tag{1}$$

$$\min R(S), R = \frac{1}{|S|^2} \sum_{x_i, x_j \in S} I(x_i, x_j). \tag{2}$$

$$\max \Phi(D,R), \Phi = D - R. \tag{3}$$

## Training the classification model

A five-fold cross-validation was adopted as mentioned before and the classifiers were trained using the training fold only. We adopted three classifiers to demonstrate the effectiveness of the chosen features. Logistic regression (*Ng & Jordan, 2002*), support vector machine (SVM) (*Cortes & Vapnik, 1995*), and random forest (RF) (*Breiman, 2001*), classifiers were used to distinguish between HGG and LGG images. The logistic classifier fits the distribution of data to the binomial distribution and provides a category related output with values between 0 and 1. The SVM is trained to maximize the margins of the plane separating the two categories in the feature space and is the most common classifier used for binary classification. RF is an ensemble classifier composed of a number of decision trees and it could lessen overfitting by training each decision tree using only a subset of the entire data. To train the logistic regression classifier, selected radiomics features were linearly regressed to binarized grades of glioma and then a radiomics score was constructed using a linear combination of regression coefficients and feature values. The score was fitted to the logistic model and a fixed threshold ($=.5$) was applied to distinguish between HGG and LGG. The cost function of the SVM was Lagrangian of the sum of the distance from each feature point to the marginal line. We chose a linear kernel for the SVM. The selected features from the feature selection step were vectorized and referred to as the 5D feature space. As a result, training of the SVM was performed using the 5D feature space. The RF model was also trained using the same information as the SVM and 200 decision trees were used for consensus result in the training step.

## Applying the model to the test cohort and statistics

We applied the trained models to the test cohort. The selected features and the associated coefficients from the training step were applied to the test cohort. The actual values of the features were computed from the test cohort. Performance of both cohorts was measured using the accuracy, sensitivity, specificity, and area under the curve (AUC) value of the receiver operating characteristic (ROC) curve. The positive case for the confusion matrix was set to HGG. The adjusted R-squared value and *p*-value were calculated to evaluate the degree of fit of the model. As we adopted a five-fold cross-validation, we repeated the procedures of feature selection, model training, and testing steps five times each time leaving a different test fold out. Each left out fold led to one set of performance measures, and thus we reported the average value of five measures. All analyses and evaluation procedures were performed using MATLAB (Mathworks Inc. Natick, MA, USA)

# RESULTS

## Selected features from training

The top five features from the mRMR feature selection algorithm were chosen as significant radiomics features from each fold. Table 2 shows the most stable four radiomics features which were selected at least three times from the five-fold cross-validation. Regarding the category of features, the morphological property of tumor was most effective at discriminating HGG from LGG. Especially, spherical disproportion, which indicated how much the tumor shape was distorted from an ideal sphere, was found to be most valuable.

**Table 2  Selected features via mRMR based on stability over five folds.**

|   | Feature name | Modality | Category | ROI type |
|---|---|---|---|---|
| 1 | Spherical Disproportion | Shape | Shape | 1 |
| 2 | Contrast | T1c | GLCM | 2 |
| 3 | Compactness | Shape | Shape | 2 |
| 4 | Autocorrelation | FLAIR | GLCM | 2 |

**Table 3  Training performance measures using various classifiers.**

| Classifier | Accuracy | Sensitivity | Specificity | AUC | Adjusted R-squared | *p*-value |
|---|---|---|---|---|---|---|
| Logistic | 0.8895 | 0.9643 | 0.6800 | 0.9066 | 0.4877 | 8.0686e−23 |
| SVM | 0.8983 | 0.9714 | 0.6933 | 0.9135 | 0.4461 | 6.5597e−13 |
| RF | 1 | 1 | 1 | 1 | 0.9537 | 7.4280e−148 |
| Average | 0.9292 | 0.9786 | 0.7911 | 0.9400 | | |

**Notes.**
Each performance value was calculated by averaging the results of the five-fold cross validation.
SVM, support vector machine; RF, random forest; AUC, area under the curve.

The next most efficacious features were the GLCM features, which represent texture characteristic of the intra-tumoral area. Regarding the imaging modality, one was from T1 contrast enhanced image, and the other was from FLAIR. Regarding the type of ROIs, one was from ROI type I (non-enhancing tumor and necrotic region) and the other three were from ROI type II (enhancing, non-enhancing tumor and necrotic region).

### Model performance in the training step

Training performance of the three classifiers is shown in Table 3. Each performance value was calculated by averaging the five-fold cross validation results. The RF classifier showed the best training performance and three classifiers had an AUC of 0.9400 on average, which showed that the classifiers were very successful at modeling the training cohort. The accuracy, sensitivity, and specificity were measured as 0.9292, 0.9786, and 0.7911 on average. Figure 3 shows ROCs for the three classifiers for all five folds.

### Model performance in test step

Table 4 shows the results of applying the model constructed at the training stage to the test cohort. These results were obtained by fixing the image features selected by mRMR and all the model parameters. The actual feature values were computed from the test cohort. Same as the training phase, the RF classifier had the best performance with AUC 0.9213, and the average AUC of the three classifiers was 0.9030. The other performance measures were obtained in the same manner as the training phase. The average accuracy, sensitivity, and specificity were 0.8854, 0.9508 and 0.7022, respectively. Figure 4 shows ROCs for the three classifiers in the test cohort for all five folds.

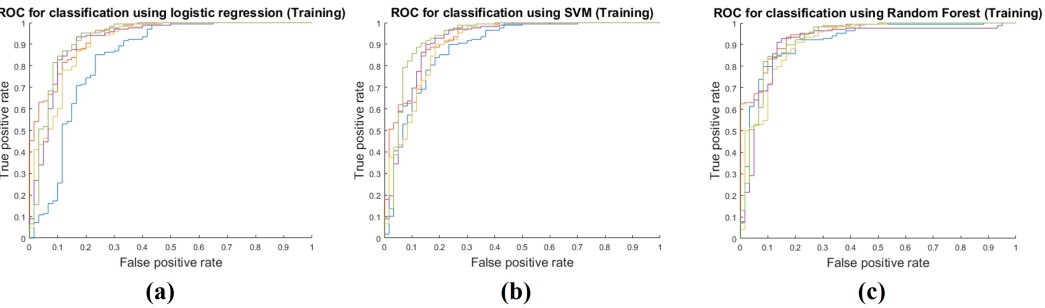

**Figure 3** **Performance curves of the five-fold cross validation in the training phase.** (A) shows the ROC for the logistic regression classifier. (B) shows the ROC for the SVM classifier. (C) shows the ROC for the RF classifier.

**Table 4** **Test performance measures using various classifiers.**

| Classifier | Accuracy | Sensitivity | Specificity | AUC | Adjusted R-squared | *p*-value |
|---|---|---|---|---|---|---|
| Logistic | 0.8877 | 0.9619 | 0.6800 | 0.9010 | 0.4882 | 5.6693e−23 |
| SVM | 0.8807 | 0.9476 | 0.6933 | 0.8866 | 0.3989 | 4.2893e−05 |
| RF | 0.8877 | 0.9429 | 0.7333 | 0.9213 | 0.5725 | 2.4653e−10 |
| Average | 0.8854 | 0.9508 | 0.7022 | 0.9030 | | |

**Notes.**
Each performance value was calculated by averaging the results of the five-fold cross validation.
SVM, support vector machine; RF, random forest; AUC, area under the curve.

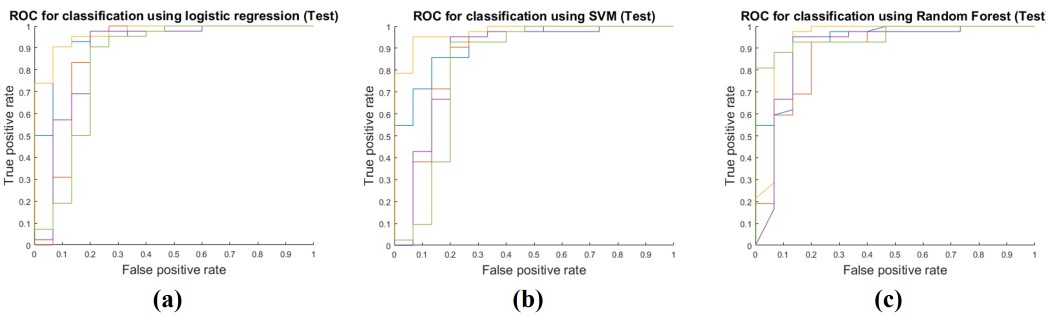

**Figure 4** **Performance curves of the five-fold cross validation in the test phase.** (A) shows the ROC for the logistic regression classifier. (B) shows the ROC for the SVM classifier. (C) shows the ROC for the RF classifier.

## DISCUSSION

A radiomics approach can compute high-dimensional features from *in vivo* imaging modalities, which in turn were used to differentiate between HGG and LGG in this study. Our radiomics approach was tested with three classifiers and we obtained a high average AUC value of 0.9030 in the test cohort. In particular, the RF classifier showed the best performance with AUC 0.9213 for the test cohort. We also tested to see if the ensemble

**Table 5  Test performance measures using various classifiers.**

| Classifier | Accuracy | Sensitivity | Specificity | AUC | Adjusted R-squared | *p*-value |
|---|---|---|---|---|---|---|
| Logistic | 0.8877 | 0.9619 | 0.6800 | 0.9010 | 0.4882 | 5.6693e–23 |
| SVM | 0.8807 | 0.9476 | 0.6933 | 0.8866 | 0.3989 | 4.2893e–05 |
| RF | 0.8877 | 0.9429 | 0.7333 | 0.9213 | 0.5725 | 2.4653e–10 |
| Ensemble | 0.8947 | 0.9571 | 0.7200 | 0.8765 | 0.5471 | 2.2992e–09 |

**Notes.**
Each performance value was calculated by averaging the results of the five-fold cross validation.
SVM, support vector machine; RF, random forest; Ensemble, ensembled classifier of three classifier; AUC, area under the curve.

of the three classifiers using majority voting could be better than individual classifiers (Table 5). The RF classifier was significantly better other classifiers and thus the ensemble procedure did not improve the overall performance. Our main contribution was to fully leverage available multi-modal imaging with various machine learning approaches to distinguish between LGG and HGG.

Others also attempted to differentiate HGG from LGG. Law et al. used relative cerebral blood volume measurements and metabolite ratios from proton MR spectroscopy, which resulted in a sensitivity of 0.950 and a specificity of 0.575 in discrimination between HGG and LGG (*Law et al., 2003*). A recent study applied conventional image processing approaches to a small scale data of 42 patients' processed perfusion MRI and achieved a sensitivity of 0.966, specificity of 0.812 and AUC of 0.95 (*Togao et al., 2016*). Zacharaki et al. adopted texture features similar as our approach to differentiate between LGG and HGG (*Zacharaki et al., 2009*). They achieved an accuracy of 0.878 and AUC of 0.896 using SVM based recursive feature elimination (SVM-RFE) with a leave-one-out cross validation. The leave-one-out cross validation portion of that study might lead to overfitting, while our study could reduce the overfitting issue with a five-fold cross validation. In addition, SVM-RFE approach is tailored for the SVM classifier and thus applying the SVM-RFE in conjunction with other classifiers could lead to performance degradation.

We found four significant features that were stable through the five-fold cross validation. They were spherical disproportion of ROI type I (non-enhancing tumor and necrotic region), contrast of GLCM of ROI type II (enhancing, non-enhancing tumor, and necrotic region) from T1-contrast enhanced images, compactness of ROI type II, and autocorrelation of GLCM of ROI type II from FLAIR images (Table 2). The spherical disproportion and compactness measure how much an ROI shape differs from a sphere. The former is a measure based on volume measurements, while the latter is based on surface measurements. For spherical disproportion, an ideal sphere has value one and the value increases as the shape differs from the sphere. For compactness, an ideal sphere has value 0.0531 (i.e.,$1/6\,\pi$) and the value decreases as the shape differs from the sphere. Glioma shape is a well-known factor associated with malignancy, as irregular tumor shape is often associated with higher malignancy and poor prognosis (*Claes, Idema & Wesseling, 2007*). We found that the shape of both non-enhancing and enhancing portion of the tumor were important in determining the glioma grades. Another important predictor of

tumor prognosis is intratumoral heterogeneity (*McGranahan & Swanton, 2015*). We found two significant texture features: the contrast and autocorrelation of GLCM. The texture features quantify textural information within the ROI and can reveal patterns of intensity heterogeneity. The contrast of GLCM measures the local intensity variation of GLCM and autocorrelation of GLCM measures the magnitude of the fineness and coarseness of textural patterns. These texture features have often been identified as significant in other radiomics studies (*Tixier et al., 2011*; *Davnall et al., 2012*; *Ganeshan et al., 2013*; *Grove et al., 2015*; *Bowen et al., 2017*). One feature was from ROI type I (non-enhancing tumor and necrotic region) and the other three were from ROI type II, which included ROI type I plus the enhancing compartment. This confirmed that we need to consider both tumor core and the enhanced portion to evaluate the tumor grading.

There are related studies of gliomas using machine learning approaches. Kickingereder et al. estimated the progression-free and overall survival of GBM patients using T1, T1-contrast enhanced and FLAIR images (*Kickingereder et al., 2016*). They used principal component analysis (PCA) to develop radiomics signatures from high-dimensional features. PCA is effective at reducing dimensionality, but its results are difficult to interpret. Itakura et al. computed quantitative features from T1 images of GBM patients and found phenotypic clusters associated with molecular pathway activity through consensus clustering (*Itakura et al., 2015*). The adopted clustering was effective at demonstrating the association between imaging features and degree of malignancy, but the results were still difficult to interpret. One study developed machine learning-based prognostic imaging biomarkers of GBM images using multi-modal imaging, similar to our study (*Cui et al., 2016*). They adopted the L1-norm regularization method to select significant imaging features and predict overall survival (*Tibshirani, 1996*). In summary, our study was designed to produce stable and interpretable results of radiomics analysis compared to existing ones.

Recently, a machine learning algorithm known as deep learning (DL) has become the go-to methodology to drastically enhance the performance of existing machine learning techniques (*Lecun, Bengio & Hinton, 2015*). DL approaches have shown promise in tumor grading, diagnosis and prognosis prediction (*Ertosun & Rubin, 2015*; *Litjens et al., 2016*; *Lao et al., 2017*; *Li et al., 2017*). DL approach does not require the researcher to specify a set of features a priori, but can implicitly learn the features relevant to the problem, and thus can be effective for radiomics research. DL requires many more training samples compared to conventional machine learning approaches and additional issues arise when fine-tuning many hyper-parameters. These issues are challenging and we plan to pursue DL approaches in the future.

Our study has several limitations. We used open source data originally designated for a segmentation challenge, so we could not control for all factors between LGG and HGG groups. This might have included bias in patient selection. Independent validation using data from another clinical site is missing. This might hinder the applicability of our approach to new data. There was a class imbalance between two classes. We thought that each class had enough samples for statistical modeling. Still, the class imbalance issue might be alleviated by minority class oversampling techniques. The ROIs were provided by the database and the reproducibility of the ROI was not verified. World Health

Organization recently announced a new tumor classification system of the central nervous system (*Louis et al., 2016*). It breaks up the glioma into five grades considering not only histological information but also isocitrate dehydrogenase mutation and 1p/19q codeletion. Updated grading of gliomas was unavailable to us and thus we used the information of the traditional grading system. Future studies should consider grading information on the new grading system.

## CONCLUSIONS

In conclusion, we showed that glioma grades could be accurately determined by a combination of high dimensional imaging features, an advanced feature selection method and machine learning classifiers. We believe the algorithm presented in our study might contribute to high-throughput computer aided diagnosis system for gliomas.

## ACKNOWLEDGEMENTS

We would like to thank the organizers of the MICCAI Brain Tumor Segmentation 2017 Challenge.

### Funding

This study was funded by the Institute for Basic Science (Grant no. IBS-R015-D1), the National Research Foundation of Korea (Grant no. NRF-2016R1A2B4008545) and the Ministry of Science and ICT of Korea under the ITRC Program (Grant no. IITP-2018-0-01798). The funders had no role in study design, data collection and analysis, decision to publish, or preparation of the manuscript.

### Grant Disclosures

The following grant information was disclosed by the authors:
Institute for Basic Science: IBS-R015-D1.
National Research Foundation of Korea: NRF-2016R1A2B4008545.
Ministry of Science and ICT of Korea: IITP-2018-0-01798.

### Competing Interests

The authors declare there are no competing interests.

### Author Contributions

- Hwan-ho Cho conceived and designed the experiments, performed the experiments, analyzed the data, contributed reagents/materials/analysis tools, prepared figures and/or tables, authored or reviewed drafts of the paper, approved the final draft.
- Seung-hak Lee performed the experiments, authored or reviewed drafts of the paper.
- Jonghoon Kim analyzed the data, authored or reviewed drafts of the paper.
- Hyunjin Park conceived and designed the experiments, analyzed the data, contributed reagents/materials/analysis tools, authored or reviewed drafts of the paper, approved the final draft.

## Human Ethics

The following information was supplied relating to ethical approvals (i.e., approving body and any reference numbers):

The institutional review broad (IRB) of Sungkyunkwan University approved our study (IRB# 2015-09-007). Consent was waived for this retrospective study. Our study was performed in full accordance with local IRB guidelines.

## Data Availability

We considered data from the MICCAI Brain Tumor Segmentation 2017 Challenge (BraTS 2017): https://www.med.upenn.edu/sbia/brats2017.html.

## Supplemental Information

Supplemental information for this article can be found online at http://dx.doi.org/10.7717/peerj.5982#supplemental-information.

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
