# Peer review of "Classification of the glioma grading using radiomics analysis"

_PeerJ, doi:10.7717/peerj.5982_

## Round 0.1 · original submission · Minor Revisions

Please address the comments of the two good reviews. In addition, explain why you achieve an accuracy of 0.88 while the Brats challenge leaderboard shows much lower scores.

Reviewer 1 ·

Basic reporting

- Clear and unambiguous, professional English used throughout.
OK

- Literature references, sufficient field background/context provided.
No, there is much more literature in survival prediction of glioma. Also the authors touch upon the literature beyond grading and survival prediction, but they do not mention anything about the field of radiogenomics, which is one of the most significant directions of our field.

- Professional article structure, figs, tables. Raw data shared.
OK

- Self-contained with relevant results to hypotheses.
Yes

Experimental design

- Original primary research within Aims and Scope of the journal.
Yes

Research question well defined, relevant & meaningful. It is stated how research fills an identified knowledge gap.
- Yes

Rigorous investigation performed to a high technical & ethical standard.
- No, there are a lot of missing (or insufficiently described) information.

Methods described with sufficient detail & information to replicate.
- No, there are a lot of missing (or insufficiently described) information.

Validity of the findings

- Impact and novelty not assessed. Negative/inconclusive results accepted. Meaningful replication encouraged where rationale & benefit to literature is clearly stated.
OK

- Data is robust, statistically sound, & controlled.
There are missing (or insufficiently described) information.

- Conclusion are well stated, linked to original research question & limited to supporting results.
Yes

Additional comments

This is an interesting manuscript, but a lot of information is missing, or insufficiently described. Before I can support its publication, the following points should be addressed

- Introduction, L40: Glioma grading is not defined purely based on histology any more. WHO has a revision for the classification of CNS tumors in 2016.

- Materials and Methods, L82: Looking into the BraTS 2017 website (https://www.med.upenn.edu/sbia/brats2017/data.html), it seems that the BraTS 2017 dataset has 4 citations that should be cited, whereas the authors include only 2 of them. All 4 citations should be cited in the paper.

- Materials and Methods, L84: According to the TCIA website, the TCIA-GBM and TCIA-LGG have one data citation each that the authors have not included I their manuscript. I include them below for the convenience of the authors. These can also be found through the official TCIA website.
TCIA-GBM Data Citation: Scarpace, L., et al. (2016). Radiology Data from The Cancer Genome Atlas Glioblastoma Multiforme [TCGA-GBM] collection. The Cancer Imaging Archive. http://doi.org/10.7937/K9/TCIA.2016.RNYFUYE9
TCIA-LGG Data Citation: Pedano, N., et al. (2016). Radiology Data from The Cancer Genome Atlas Low Grade Glioma [TCGA-LGG] collection. The Cancer Imaging Archive. http://doi.org/10.7937/K9/TCIA.2016.L4LTD3TK

- Materials and Methods, L87: According to the BraTS 2017 documentation the HGG cohort includes only glioblastoma tumors. The authors correct their statement.

- Materials and Methods, L110: Are these features extracted from the 3D volume, or in 2D from the slice with the largest expansion of the tumor?

- Materials and Methods, L115: A lot of details are missing both from the main paper and the supplementary material for the feature extraction to be reproducible. The authors should clarify the parameterization for each feature family. For example, what was the quantization method used to extract the GLCM features? How many bins were selected? How many offsets, directions were used for GLCM? How the different offsets were combined? by weighted average, or just a combination?

- Materials and Methods, L124: Has the z-scoring happened for each feature across subjects, or across features for each subject?

- Materials and Methods, L143: This “5D feature space” was not mentioned before. The authors should elaborate how it is defined. How was the SVM trained did the authors cross-validate for “c” and “g” parameters?

- Materials and Methods: The authors do not discuss at all about their divided cohort. Is the 5-fold created proportionally to the number of subjects of each class (high grade vs low grade)? What is the ratio of each class in relation to the training vs testing dataset? What is the ratio of overall subjects between the training and the testing datasets?

- Materials and Methods: Since the authors already have results from the 3 algorithms they applied, it would be very interesting to assess the classification performance of an ensemble across the 3 methods.

- Discussion, L214:: The authors should elaborate and clarify on how (and why) the SVM-RFE with LOOCV might lead to overfitting.

- Table 1 seems to be taken from the Bakas, et al. 2017 Sci Data paper. This should be mentioned in the caption of the table.

·

Basic reporting

It was a pleasure to review the manuscript titled "Classification of the glioma grading using radiomics analysis". The authors focus on applying various machine learning classification algorithms to a specific problem such as the classification of grading in gliomal brain tumours. The problem statement has been clearly defined.

However, the methods and results sections would be more helpful to prospective readers if a few things can be clarified or added.

Experimental design

Methods:
1. There is evidently a class imbalance present in the cohort, with 210 HGG cases and only 75 LGGs. Has this been addressed or if the authors did not deem this to be a significant issue, can they explain why?
2. Were there any other methods employed to reduce model overfitting like elasticnet regression or lasso? The relatively high AUCs suggest overfitting.
3. The 'Tumour regions of interest' is not clear in explaining why the regions were combined. Also, the second column in Figure 2 that depicts radiomics feature extraction needs more explanation as in it's current state, the figure is not able to provide a clear understanding of the methodology.
4. The reason behind normalizing features to z-scores should be explained.
5. Is there a reason for selecting the 3 classifiers mentioned in the article? Additionally, the relative merits and demerits of each of those classification algorithms can be mentioned in the article, or in the least a reference(s) should be provided.
6. Is there a reason for selecting only the top 5 features from training?

Validity of the findings

Results:
7. All the tables are titled Table 1.
8. Figure 2 warrants more explanation.
9. The supplemental information provides useful information regarding the variables. Some representative examples of the results of morphological features can be included for completeness.
10. The robustness of the data cannot be commented upon, since it comes from an external source.
11. The conclusions are well explained and address the original research question stated at the beginning of the article.

Additional comments

Overall, this is a well-written paper and with the additions suggested above, can be a good reference for future researchers interetsed in exploring machine learning methodologies in the computer aided diagnosis and digital pathology domains.

---

## Round 0.2 · accepted · Accept

I agree with the reviewers that your rebuttal and updated manuscript addressed all comments.

Please address the minor remaining comment from reviewer 1 (adding the response table). You can do this while in Production

Reviewer 1 ·

Basic reporting

Clear and unambiguous, professional English used throughout.
ok

Literature references, sufficient field background/context provided.
ok

Professional article structure, figs, tables. Raw data shared.
ok - Please note that the authors should include the "response Table 1" into the main body of the manuscript.

Self-contained with relevant results to hypotheses.
ok

Experimental design

Original primary research within Aims and Scope of the journal.
ok

Research question well defined, relevant & meaningful. It is stated how research fills an identified knowledge gap.
ok

Rigorous investigation performed to a high technical & ethical standard.
ok

Methods described with sufficient detail & information to replicate.
ok

Validity of the findings

Impact and novelty not assessed. Negative/inconclusive results accepted. Meaningful replication encouraged where rationale & benefit to literature is clearly stated.
ok

Data is robust, statistically sound, & controlled.
ok

Conclusion are well stated, linked to original research question & limited to supporting results.
ok

Speculation is welcome, but should be identified as such.
ok

Additional comments

The authors have sufficiently addressed my comments.

My only minor suggestion to the authors would be to include the "Response Table 1" ("Response 15") in the main body of their manuscript.

·

Basic reporting

This is a review for a revised version of the manuscript. The newer version meets expected standards.

Experimental design

The changes that were suggested in the original review have been incorporated satisfactorily. Commendations to the authors for performing these further investigations that will definitely help future readers.

Validity of the findings

The authors have done a thorough job of carrying out experiments to address some of the questions from the original review. The results, especially the graphic representations that I had requested for, make the paper more understandable and interpretable.

Additional comments

Thank you for carrying out the changes as requested previously.